# Assessing the Impacts of Creating Active Schools on Organisational Culture for Physical Activity

**DOI:** 10.3390/ijerph192416950

**Published:** 2022-12-16

**Authors:** Zoe E. Helme, Jade L. Morris, Joanna Nichols, Anna E. Chalkley, Daniel D. Bingham, Gabriella M. McLoughlin, John B. Bartholomew, Andrew Daly-Smith

**Affiliations:** 1Faculties of Life Sciences and Health Studies, University of Bradford, Richmond Road, Bradford BD7 IDP, UK; 2Centre for Applied Education Research, Wolfson Centre for Applied Health Research, Bradford Royal Infirmary, Bradford BD9 6TP, UK; 3Centre for Physically Active Learning, Faculty of Education, Arts and Sports, Western Norway University of Applied Sciences, 5063 Bergen, Norway; 4Born in Bradford, Bradford Institute for Health Research, Bradford Teaching Hospitals Foundation Trust, Bradford BD9 6RJ, UK; 5College of Public Health, Temple University, Philadelphia, PA 19140, USA; 6Implementation Science Center for Cancer Control and Prevention Research Center, Brown School, Washington University, St. Louis, MO 63130, USA; 7Department of Kinesiology and Health Education, The University of Texas at Austin, Austin, TX 78712, USA

**Keywords:** physical activity, whole-school approach, organisational change, creating active schools

## Abstract

Background: National and international guidance recommends whole-school approaches to physical activity, but there are few studies assessing their effectiveness, especially at an organisational level. This study assesses the impact of the Creating Active School’s (CAS) programme on organisational changes to physical activity provision. Methods: In-school CAS leads completed a 77-item questionnaire assessing school-level organisational change. The questionnaire comprised 19 domains aligned with the CAS framework and COM-B model of behaviour change. Wilcoxon Signed Rank Tests assessed the pre-to-nine-month change. Results: >70% of schools (*n* = 53) pre-CAS had inadequate whole-school physical activity provision. After nine months (*n* = 32), CAS had a significant positive effect on organisational physical activity. The positive change was observed for: whole-school culture and ethos, teachers and wider school staff, academic lessons, physical education (PE) lessons, commute to/from school and stakeholder behaviour. Conclusions: This study provides preliminary evidence that CAS is a viable model to facilitate system-level change for physical activity in schools located within deprived areas of a multi-ethnic city. To confirm the results, future studies are required which adopt controlled designs combined with a holistic understanding of implementation determinants and underlying mechanisms.

## 1. Introduction

Globally, only 40% of children accumulate 60 min of physical activity every day as recommended by the World Health Organization (WHO) [1,2]. Consequently, many children miss out on the multiple benefits (e.g., body composition, social skills, mental health, academic achievement, motor skills development) of regular physical activity [3]. Schools are seen as a pragmatic setting in which physical activity interventions can be delivered to a large number of children [4]. However, studies demonstrate that UK-based primary school children are active for an average of 18.33 ± 8.34 min of moderate-to-vigorous physical activity (MVPA) in school time, with 90.2% failing to achieve the 30 min in-school physical activity target [5,6]. It is therefore important to understand why attempts to increase children’s physical activity levels have been unsuccessful via school-based approaches when research and international guidance suggest that they are effective settings within which to intervene [7,8,9].

Increased emphasis on health-related policy goals for physical education (PE) and school sports from national and international governing bodies (Education Departments, WHO, International Society for Physical Activity and Health), has placed an onus on schools to identify and implement strategies that support pupils to be habitually active and work towards achieving the physical activity guidelines [8,9,10,11]. However, whilst these strategies suggest the need to increase opportunities for children to be active during school, current evidence highlights a continuing focus (and dependency) on single solutions aimed at an individual-level behaviour change or targeted populations (e.g., racial/ethnic minorities, girls) [4]. This results in the ongoing implementation of singular-level interventions that employ ‘quick-fixes’ and, thus, have little or no impact on sustainable or equitable physical activity changes [8,10].

Recently, the WHO published the Global Action Plan for Physical Activity, calling for systems-thinking to create sustainable changes to physical activity provision across the whole school [9]. Systems-thinking emphasizes the importance of understanding the ‘whole-system’, and the complex relationships among components and underpinning mechanisms [12]. Contemporary whole-school approaches, such as the Action Schools! BC model and Comprehensive School Physical Activity Programme (CSPAP) [13,14], have received attention from governing bodies and researchers given their potential to stimulate organisational change (e.g., school policy, multiple opportunity areas), to physical activity provision [15,16]. However, while behavioural theory and implementation science have been added in recent years to the aforementioned models and subsequent programmes, both were lacking in their original development [17,18]. Further, evaluations of these whole-school approaches to date have investigated the effectiveness of individual components in isolation (e.g., increased PE, active travel, active classrooms), with little consideration of the implementation and impact of a comprehensive multi-level approach with multiple stakeholders [4]. Consequently, there is a need to understand how to design and implement such interventions to achieve changes in physical activity promotion from a whole-systems perspective.

The CAS framework is the first whole-school approach to integrate multi-stakeholder perspectives with behaviour change theory (BCT) and implementation science to ensure it is informed by research and practice [19]. The framework underpinned the development of the CAS programme that focuses on transforming organisational culture to promote physical activity in schools across four domains: policy, environments, stakeholders, and opportunities. The framework incorporates both the COM-B model of behaviour change identifies three factors that must be present for behaviour change to occur [17], and the Consolidated Framework for Implementation Research (CFIR) to identify physical activity determinants across multiple levels of the organisation [20] (Appendix A). For more detail on the CAS framework and its theoretical underpinnings, please see Daly-Smith et al., 2020 [19]. This study aims to evaluate the effectiveness of the CAS programme on organisational culture for physical activity in schools. A secondary aim is to establish the internal validity and reliability of a school-based organisational capacity questionnaire.

## 2. Materials and Methods

### 2.1. Intervention Context and Description

The CAS programme focuses on the school’s assets (e.g., facilities, environment, staff, capacity) to promote behaviour change using the COM-B framework [21]—across four specific areas: policy, environments, stakeholders, and opportunity. CAS is led by a national team (University of Bradford (UoB), Yorkshire Sport Foundation (YSF), and Bradford Institute for Health Research (BIHR)) who design and implement the programme across the UK. The programme has strategic leads for research and development (UoB) and partnership development (YSF). A national manager (YSF) leads the delivery of locality leads across a national community of practice, of which the Bradford locality is a member. This study focused on Bradford-based schools, the initial location for implementation. Bradford is one of the most ethnically diverse cities in the country with 34% of its neighbourhoods in the lowest tertile of deprivation across the UK [22,23].

Three delivery models (JU:MP; Living Well RIC; CP5) for the CAS programme were trialled with Bradford Schools (Figure 1). The differences in the three programmes were due to varying funding arrangements for different city wards. Living Well RIC offered support for CAS with no additional funding to implement. JU:MP and CP5 both supplemented the support for CAS with small grants (£4K–£10K). All schools received the same implementation support from the CAS team, with the funding providing additional resources to support the schools to implement their chosen initiatives. All programmes were coordinated by the locality leadership team of the UoB and BIHR.

The national CAS team and locality leadership team trained and supported the Bradford-based CAS Champions, who in turn, supported an allocated set of local schools (~3–4). CAS Champions (locality leads) were recruited from local schools (*n* = 11) and the public health team (*n* = 3). Their role was to provide external support to schools, to facilitate engagement with CAS, check and challenge during the profiling exercise, support the development and implementation of school-based initiatives and connect schools into locality-based communities of practice. CAS Champions received two days of training with regular support webinars and face-to-face meetings throughout the year. They met with their allocated schools to facilitate CAS programme adoption, at termly (3 times a year) physical activity conferences and then at key points throughout the year as requested by the schools (~2 to 4 times). Once signed up to the CAS programme, schools started a four-stage annual CAS cycle beginning with a self-assessment. For this study, the self-assessment began in October/December of 2021. These four stages are outlined as follows:

Stage one (October to December 2021): the CAS Champion supported in-school CAS leads to complete an online profile assessment (creatingactiveschools.org) of whole-school physical activity provision aligning with the four CAS framework areas: policy (includes 5 domains), environments (5 domains), stakeholders (5 domains), and opportunities (7 domains). After completing the profile, schools received automated summary scores and recommended priority actions based on areas for high impact. CAS Champions received a summary report for the schools within their locality.

Stage two (November 2021 to January 2022): schools identified a maximum of three priority areas from the 22 domains. Priorities were integrated into the School Development Plan for the current academic year (2021/2022). The in-school CAS lead completed a Planning for Change document using quality assurance criteria (APEASE [21]) to identify evidence-informed physical activity initiatives to address the priority areas [21]. The CAS Champion identified opportunities for shared physical activity initiatives across their allotted schools, informed by the data in the profile tool locality dashboard. The CAS Champion created face-to-face communities of practice amongst their allocated schools to develop and support the implementation of shared school initiatives (e.g., integrating physical activity within teaching and learning policy, developing active travel initiatives, and outdoor physically active learning opportunities). This involved tendering for external support and identifying pioneering schools or in-school leads to support other schools around specific agendas (e.g., PE or outdoor learning).

Stage three (December 2021 to July 2022): The CAS Champion supported the implementation of individual and collective school initiatives. The locality leadership organised termly conferences/communities of practice (3 times a year) to support the development and implementation of initiatives, these included; policy (e.g., integrating physical activity in teaching policy) or environment improvements (e.g., building an outdoor classroom), stakeholder training (e.g., physically active learning continuing professional development), and direct support for physical activity within one of the seven CAS opportunity areas for physical activity (e.g., bikeability training to support active travel).

Stage four (December 2021 to July 2022): Schools were encouraged to monitor the impact of their initiatives through Sport England Active Lives Survey (www.sportengland.org/research-and-data/data/active-lives, accessed on 12 December 2022) and/or in-school surveys/focus groups with staff and children. Some schools bought devices (e.g., pedometers) to assess changes in physical activity. This data was used to inform the next CAS profiling exercise, starting a new annual cycle for the 2022/2023 academic year.

### 2.2. Study Design

A longitudinal mixed-method evaluation was designed to measure the effectiveness of implementing CAS within primary schools in Bradford. The current study draws on the assessment of organisational change for physical activity provision over the first nine months of the evaluation. See Appendix A for further detail on the alignment of this study within the overall evaluation of the CAS programme in Bradford.

### 2.3. Sampling and Participants

All primary schools in the wards aligned to JU:MP, Living Well RIC and CP5 programmes were invited to take part in the CAS programme, commencing September 2021 (*n* = 57). Of these, 53 (92.98%) schools volunteered to participate in the current study. This involved the in-school CAS lead participating on behalf of the school. All participants received a study information sheet including instructions for completion prior to completing an organisational change questionnaire. Consent was gained upon submission of baseline and follow-up questionnaires. Ethical clearance was granted by the UoB Research and Innovation Services (E926).

### 2.4. Data Collection

School readiness and organisational capacity for physical activity were assessed over a nine-month period via a questionnaire that the in-school CAS leads completed on behalf of their school and stakeholders. Baseline measurements were conducted (October–December 2021) prior to schools engaging with the CAS programme, and again at a nine-month follow-up (July–September 2022). At both timepoints, a paper version of the questionnaire was given at the face-to-face CAS conference/community of practice event. A link to an online version was then sent to in-school CAS leads from the schools not in attendance. Email reminders to complete the questionnaire were sent after two and four weeks. CAS Champions were also asked to prompt schools to complete questionnaires.

### 2.5. Measures

The organisational change questionnaire was developed through a multistage consultation to increase ecological validity and ensure language and terminology would be easily understood by those with less experience in physical activity programming. The research team sought to ensure questions aligned with the COM-B and CAS frameworks. First, the paper authors who have expertise in school-based physical activity, behaviour change, and implementation science reviewed the previously validated school wellness questionnaire (SWQ) [24], CAS framework [19], CAS profiling tool, COM-B model [21], and six-item COM-B questionnaire assessing the effectiveness of interventions [25]. From this, a set of questions were developed that focused on school readiness and organisational capacity to promote physical activity. A draft questionnaire was then reviewed by a practising head teacher and an independent academic with expertise in evaluating behaviour change. Both provided feedback on questions’ context and structure following amendment, the questionnaire was then trialled with the national CAS team and Bradford-based CAS Champions (*n* = 12) before final amendments were made to the structure and question-wording. Amendments across both stages of feedback specifically focused on simplifying the behavioural terminology for school stakeholders.

#### Organisational Change Questionnaire

The questionnaire measured organisational provision for physical activity across five domains: whole-school culture and ethos, communication, physical environments, stakeholders, and physical activity opportunities (full questionnaire, Appendix A). In total there were 77 questions all scored using a five-point Likert-type scale (strongly disagree to strongly agree). The specific focus of each domain is summarised below:Whole-School Culture and Ethos: questions relating to policies, strategies, monitoring and evaluation, and evidence-based practice for physical activity across the school (9 questions).Communication: questions on both internal (within school) and external (outside of school) communications for physical activity (2 questions).Physical Environments: questions on each environment within the school that enables physical activity (7 questions).Stakeholders: questions relating to the provision of physical activity and behaviours of each stakeholder group identified on the CAS framework [19] (21 questions).Physical Activity Opportunities: questions referring to the provision of physical activity within each of the CAS framework’s seven opportunity areas [19] (38 questions).

A summative understanding of stakeholder behaviour was generated by collating the scores from individual questions across the five domains aligned with capability (19 questions), opportunity (31 questions), and motivation (8 questions) [17,21]. Appendix A shows specific questions aligned with the COM-B model.

### 2.6. Data Analysis

Data were included if an in-school CAS lead had completed the questionnaire on behalf of their school at both time points. All data were analysed in STATA (Version 17.0) (Stata Corp, College Station, TX, USA). First, Cronbach’s Alpha was used to assess the internal consistency of the whole questionnaire, each domain and individual question due to it being a bespoke measure for organisational change [26]. Individual items were included if they achieved an items-rest correlation score of α ≥ 0.3 [27]. The questionnaire and each domain were accepted as worthy of retention if a score of α ≥ 0.7 was achieved [26]. If individual items scored α < 0.3 they were removed from the analysis and the domain was re-assessed. After review, some items achieving scores of α < 0.3 were still included in the analysis as an individual item rather than a cluster if the domain was deemed to be measuring different constructs. COM-B domains were also assessed for internal consistency using the same procedure.

For baseline data (*n* = 53), descriptive statistics (median and frequency of response) were calculated for all domains. Aligning with the Likert-scale scoring, median scores of ≤3 (neutral, disagree, strongly disagree) were perceived as domains in which physical activity provision was inadequate, whilst domains with median scores of ≥4 (agree, strongly agree) were suggested to have adequate physical activity provision.

Next, descriptive statistics, the median and interquartile range (IQR), were calculated for all domains for schools that completed baseline and follow-up questionnaires (*n* = 32). As the data were ordinal, non-parametric and not normally distributed, the Wilcoxon Signed Rank Test was performed to assess the difference in physical activity provision at an organisational level from when schools joined the programme to the nine-month engagement point. Statistical significance was defined as *p* < 0.05 [28]. The effect sizes of each domain were interpreted as small (0.10), medium (0.30), or large (0.50) [29]. The same procedure was conducted for the COM-B domains. Factor analysis was not performed due to (a) the questionnaire within this study being developed from previously validated questionnaires, and (b) the sample size being deemed too small, and rather the reliability (Cronbach Alpha) was more suitable for this study [26,30].

## 3. Results

### 3.1. Sample Characteristics

Fifty-three in-school CAS leads completed the questionnaire at baseline (Table 1). Across all delivery methods, a total of 32 (60.38%) in-school CAS leads completed the questionnaire at both timepoints and were included in the analysis (Table 1). There were no discernible patterns in those who completed the follow-up questionnaire as a function of their programme (JU:MP; Living Well RIC; CP5). Three-quarters of the schools included in the final sample lie within the top 10% of the most deprived neighbourhoods in the UK.

### 3.2. Reliability of the Questionnaire

Cronbach’s Alpha showed the questionnaire reached acceptable reliability (α = 0.92). Fifteen of the 19 domains were retained, with the exceptions being communication (α = 0.57), senior leaders (α = 0.42), children (α = 0.55), and events/visits (α = 0.29). Inter-item correlations showed specific questions impacting domain consistency, these were removed prior to the main analysis. Questions within the communication, children and events/visits domains were included but treated as individual items rather than a cluster due to each item measuring different constructs. Following the removal of the questions, repeated Cronbach’s Alpha showed acceptable reliability for the whole questionnaire (α = 0.93), and individual domains. Stakeholder capability (α = 0.91), opportunity (α = 0.86), and motivation (α = 0.84), all achieved acceptable reliability.

### 3.3. Baseline Descriptives

At the baseline (Figure 2), 72% of schools rated their whole-school physical activity culture and ethos as inadequate (≤3). Both internal (79%) and external (70%) communication was also rated as inadequate by most schools. Two-thirds (64%) of schools rated the physical environment as adequately influencing physical activity provision (≥4). Results varied across stakeholder groups. The senior leader domain was ranked highest with 92% of schools rating support as adequate. Conversely, the parents’ domain (96%) and community stakeholders’ domain (81%) were predominantly perceived as inadequate for most schools. Evidencing variability across schools, fifty-one rated teachers and wider school staff as inadequate. The role of children varied with only 26% of schools providing adequate opportunity for children to be involved in whole-school physical activity provision, while influence (57%) and peer support (51%) were more common across schools.

School provision for physical activity opportunities varied considerably across categories. Three-quarters of schools rated the provision of physical activity in PE (75%), break/lunch (75%), events/visits: external sites (81%), commute to/from school (87%), and family/community (81%) as insufficient. There was greater variability between schools in the provision of physical activity in academic lessons and Events/Visits: within/between schools as 49% and 55% of in-school CAS leads, respectively, rated this as adequate. The only opportunity for physical activity that was rated adequate by most schools (72%) was before/after school clubs.

### 3.4. Organisational Change for Physical Activity Provision

Wilcoxon Signed Rank analysis revealed a large, significant effect of the CAS programme on organisational change for physical activity (Table 2). Across questionnaire domains, large effect sizes were observed for the intervention for whole-school culture and ethos, teachers and wider school staff, academic lessons, PE lessons, and commute to/from school. Significant, medium intervention effects were detected for children: volunteer opportunities and before/after school clubs. Seven domains showed no change from baseline to follow-up.

### 3.5. Behaviour Change

Significant effects were seen for all measures of behaviour (Table 3). Large effect sizes were seen for both Stakeholder Capability and Opportunity, and only medium effect for Stakeholder Motivation (Table 3).

## 4. Discussion

The current study assessed the effect of the novel CAS programme on organisational change and stakeholder provision for physical activity. Following nine months of CAS implementation, there were large positive effects on the organisational physical activity culture of schools. Large effects were also observed for whole-school culture and ethos, a measure of the schools’ policies, strategies, monitoring and evaluation, and evidence-based practice around physical activity. Overall, there were large significant effects of CAS on Stakeholder Capability and Opportunity, and medium effects on Stakeholder Motivation to adopt and implement a whole-school approach to physical activity. Whilst the senior leader domain scored highly across time points, there was a significant positive change in teachers’ and wider school staff responses, indicating that they were the initial beneficiaries of the CAS programme. Parent behaviour—the lowest domain score pre-intervention—showed no improvement indicating parents were not reached in the initial stages of CAS implementation. At the opportunity level, large effects were observed for physical activity provision during academic lessons, PE, and commute to/from school, suggesting that schools focus on these areas in the early stages of CAS.

Given that most of these schools received at least some minimal funding (22 of the 32 in the final sample), many were advantaged in their ability to implement institutional change for physical activity. Interestingly, there was no difference in the pattern of changes implemented between those schools (*n* = 22) and those without funding (*n* = 10). Consequently, the results from this study indicate that the CAS programme is broadly successful in guiding the development of opportunities for physical activity in schools. These findings contribute to the growing evidence base surrounding whole-school approaches to physical activity, reinforcing the use of CFIR and other implementation frameworks and strategies to create organisational change in schools for physical activity provision [31]. In alignment with the SWITCH programme—a whole-school approach designed to support school wellness and reduce youth obesity levels—the results showed significant changes to the provision of opportunities for physical activity at an organisational level [24,31]. Results may be similar due to the SWITCH measure of organisational capacity underpinning the current questionnaire [24]. While an initial change has been detected at nine months of CAS and implementation support, future research needs to track the longevity of change and investigate the underlying mechanisms of change and the contextual factors that likely influence these.

Results suggest CAS improves the organisation-level (school-based) physical activity strategies, policies, monitoring and evaluation, and evidence-based practice. Such changes are essential as policy, culture, and ethos drive systemic change throughout an organisation [19]. Previous studies examining the impact of school-based policies to promote healthy behaviour have demonstrated a direct influence on physical activity areas [15,32].

The mechanisms by which policy and higher system improvements lead to behaviour change at the pupil level, are through enhanced social and physical environments and improved stakeholder behaviour to adopt and implement physical activity [19,33]. In combination with previous work [33,34], the results address the need for national guidance to recommend whole-school approaches rather than what has often been ineffective single-component or multi-component interventions.

The results suggest that the use of the CAS programme improves the capability, opportunity, and motivation of school-based stakeholders to adopt and implement physical activity [21]. As seen in a recent meta-synthesis, changes in all three domains of behaviour are essential to improve physical activity delivery by teachers and wider school staff [35]. The results of this study provide novel insights into how stakeholder behaviour can positively influence broader whole-school culture for physical activity. Findings should, however, be considered cautiously. This study was limited to the initial adoption of opportunities for physical activity in schools. It did not include an evaluation of the actual student physical activity, nor the maintenance of any systems change. As a result, the improvements may reflect a novelty to engage in an initiative and further work is required to track changes over the longer term [35], along with an actual change in pupil physical activity behaviours.

Surprisingly, changes in behaviour were not observed in school leaders. This may be explained by high pre-intervention scores and that middle or senior leaders were involved in the completion of the questionnaire. Future research is warranted to better understand the role of senior leaders in the adoption and implementation of CAS due to their influence on school priorities [4], and their support is shown as highly correlated with the adoption of opportunities for physical activity in schools [36]. This might be best achieved through qualitative work to understand the complexity and perceived compatibility and relative advantage of CAS as these factors can be a barrier to senior leaders adopting and maintaining practices at a school-wide level [4,37].

Teachers and wider school staff were the only group to show a significant positive behavioural change in the initial nine-month period. Contemporary evidence suggests that successful implementation is dependent on teachers as the key agents who deliver interventions [35]. With the most direct influence on children throughout the school day, teacher buy-in is critical to the programme’s success [4,35]. Similar to the Finnish Schools on the Move programme, CAS empowers school staff to adopt and maintain new practices in individualized ways (e.g., selecting initiatives based on current school assets and contexts) [33,34]. Such approaches aim to support longer-term behaviour engagement as teachers can flex programme delivery to meet their unique needs and the dynamic nature of the school environment. Similar to the SWITCH intervention [31], parents’ behaviour scored lowest pre-intervention and did not improve over the initial nine-month period, suggesting that CAS did not engage parents within the early stages of the intervention. These findings are consistent with the SWITCH programme where parents had low levels of engagement due to reduced representation within the programme [24]. Moving forward, CAS and other whole-school approaches to physical activity need to develop effective parent engagement strategies to support parents in promoting and providing physical activity opportunities throughout the school day and beyond as they are the key agents in facilitating children’s health behaviours [38].

At the opportunity level, CAS was shown to have large significant effects on academic lessons, PE, and the commute to/from school. This is not surprising given the behaviour change largely centred on teachers who have greater control over PE and academic lessons [4]. In addition, these areas have less barriers than environmental or broader policy change [35]. This is quite clear with the integration of physical activity into academic lessons. The appetite to focus on introducing physical activity into academic lessons aligns with the prioritization of this opportunity being placed at the centre of the CAS framework and agrees with previous whole-school programmes [39]. This is consistent throughout literature as curriculum-based initiatives appear more feasible than the environment- or policy-related practices [35,39]. Further, this could be influenced by the strong body of research that links physically active learning and classroom movement breaks to increases in cognition and academic performance [40,41]—a current priority as schools continue to rebuild following the COVID-19 pandemic. Connecting the system, there is likely a symbiotic relationship between teacher behaviour to adopt and implement physical activity and an increased use of physical activity within academic lessons. In addition, schools in Bradford developed a community of practice around PE, which could explain why this improved over the initial nine months, compared to other opportunities. Similarly, CAS within Bradford is part of a whole-system change to physical activity for children and young people; active travel across the community is one of the central work packages, with schools receiving additional support. However, this study cannot determine why change has been seen, reinforcing the use of future qualitative research to aid a more holistic understanding of the implementation and identify potential mechanisms causing change.

### 4.1. Future Directions

This study was designed to determine the impact of the CAS programme on short-term organisational change for physical activity in schools. Future work needs to assess the sustainability of these changes and future effects to increase policies, environments, stakeholder behaviour, and opportunities through the CAS programme. This requires a longitudinal follow-up across the length of time predicted for the transformation of school culture (years vs. months). However, denser sampling methods on finer timescales are also warranted to examine the causal mechanisms behind the change and the interactions across the multiple levels of the school system. Additionally, a qualitative measure should be added to gain a holistic view as to where and why changes have been seen, and to include the perspectives of multiple stakeholders. Moreover, aligning with research, schools should invest in the development of a comprehensive physical activity policy to be adopted throughout the school day and by all school stakeholders, especially the senior leaders, to enhance the likelihood of systematic, sustainable whole-school changes to physical activity provision. Finally, future work should measure pupil-level physical activity in combination with the assessment of organisational change. This will allow an examination of the relationship between changes at higher system levels in school and the impacts that these may have on pupil physical activity levels.

### 4.2. Strengths and Limitations

This is the first paper to evaluate the CAS programme, providing novel insights into the UK’s first whole-school approach to physical activity underpinned with behavioural and implementation science. As a result, this paper allows for inferences to be made for future testing and assessment. Reliance on one individual, who may be susceptible to priming beliefs, to assess overall organisational changes reduces the validity of findings. Therefore, observational studies would provide more objective data to corroborate self-report outcomes in relation to school practices and staff behaviours. Further, the lack of a control group reduces confidence that positive effects are due to the implementation of CAS rather than extraneous variables. This is especially true given that schools were part of a programme offering funding and/or broader whole-system support for physical activity through the JU:MP programme [42]. Future studies using controlled designs are warranted to confirm current findings. In addition, future studies using a mixed-method design and a broader range of stakeholders would improve understanding of how contextual factors and varied programme funding influences engagement and implementation of the CAS programme.

Using bespoke questionnaires to understand implementation is common practice due to the need to evaluate specific components of a programme. While previous work raises concerns with this approach due to difficulties in comparing findings, highlighting good practice, questionnaire development was informed by existing questionnaires [24,25] and included multiple stages of development and refinement that involved researchers, specialists in implementation science and educational practitioners. Further, statistical refinement via Cronbach Alpha assessed and increased the internal consistency and reliability of the questionnaire.

## 5. Conclusions

This study provides preliminary evidence that CAS is a viable model to facilitate system-level change for physical activity in schools located within deprived areas of a multi-ethnic city. Given the importance of reducing health inequality, the results are promising and warrant further examination. To increase the generalisability of findings, the study must be repeated in multiple contexts and areas and over a longer period. The longevity of change must be explored to determine further change and if the change is sustainable. Using qualitative research will enable an understanding of the implementation factors that have influenced the positive outcomes and to understand the underlying mechanisms of change.

The ability of the CAS programme to improve higher-system factors such as policies, strategies, monitoring and evaluation, and using evidence-based practice is promising for future whole-school physical activity approaches. It was interesting that teachers and wider school staff were the beneficiaries in the initial nine months. To ensure a true whole-school approach, it will be essential for CAS to work with broader stakeholders within and beyond schools (e.g., parents and public health specialists) to adopt and implement the physical activity.

## Figures and Tables

**Figure 1 ijerph-19-16950-f001:**
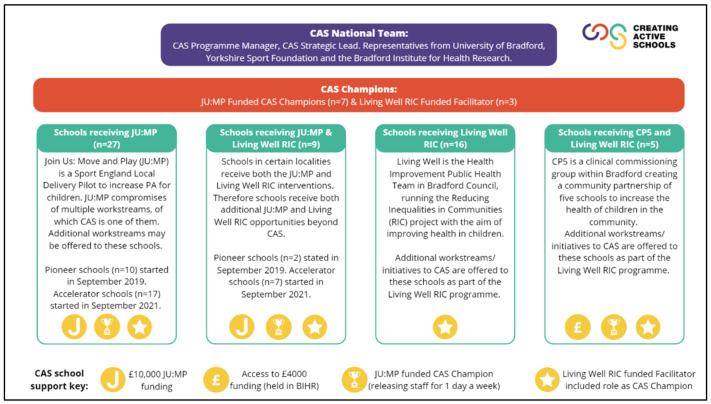
The delivery model of CAS within Bradford.

**Figure 2 ijerph-19-16950-f002:**
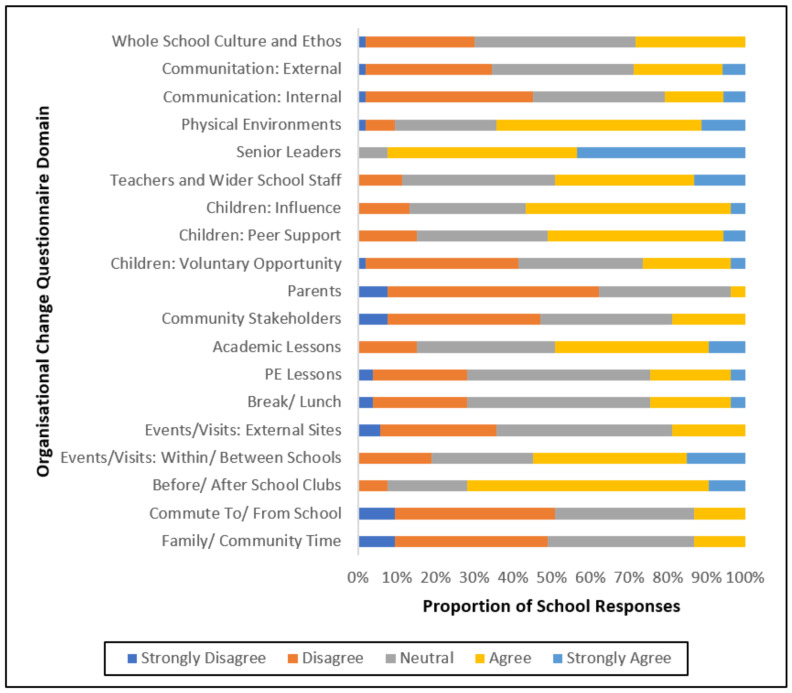
Descriptive statistics of the baseline measures of organisational changes to physical activity provision.

**Table 1 ijerph-19-16950-t001:** The number of schools in the baseline and final data sample by programme.

	JU:MP	JU:MP and Living Well RIC	Living Well RIC	CP5 and Living Well RIC
Pre data (N° of schools)	23	9	16	7
Pre and nine-month follow-up (N° of schools)	13	7	10	5

**Table 2 ijerph-19-16950-t002:** Descriptive statistics and results from the Wilcoxon Signed Rank test for changes in organisational physical activity provision.

Domain	Items	Pre		Post		Change	Z **	*p*	r ***
		Median	IQR *	Median	IQR *				
Organisational Change	77	3.1	2.8–3.2	3.3	2.9–3.6	0.2	3.35	0.0005	0.59
Whole-School Culture and Ethos	9	3	2.6–3.5	3.5	2.9–3.9	0.5	3.22	0.0013	0.57
Communication (1: External)	1	3	2–4	3	2.5–4	0	1.15	0.2789	0.21
Communication (2: Internal)	1	3	2–3	3	2–4	0	1.74	0.0926	0.31
Physical Environments	9	3.3	3–3.7	3.5	3.21–3.7	0.2	0.43	0.674	0.08
Senior Leaders	6	4	3.7–4.5	4	4–4.5	0	1.51	0.1332	0.27
Teachers and Wider School Staff	7	3.3	2.9–3.6	3.7	3.6	0.4	3.14	0.0012	0.56
Children (1: Influence)	1	4	3–4	4	3–4	0	1.22	0.236	0.22
Children (2: Peer Support)	1	3	3–4	4	3–4	1	1.53	0.1715	0.27
Children (3: Voluntary Opportunity)	1	3	2–4	4	2.5–4	1	2.44	0.0158	0.43
Parents	2	2	2–2.5	2.5	2–3	0.5	1.78	0.0807	0.31
Community Stakeholders	1	3	2–3	3	2–4	0	1.08	0.3087	0.19
Academic Lessons	6	2.6	2.2–3	3.1	2.3–3.4	0.5	3.21	0.0008	0.57
PE Lessons	6	3.2	2.7–3.6	3.7	3–4	0.5	2.92	0.0027	0.52
Break/Lunch	6	3	2.5–3.7	3	2.5–3.6	0	1.18	0.2379	0.21
Events/Visits (1: External Sites)	1	3	2–3	3	2–4	0	1.92	0.0641	0.34
Events/Visits (2: Events within/between schools)	1	3.5	3–4	4	2–4	0.5	0.20	0.9105	0.04
Before/After School Clubs	6	3.5	3.3–4	3.8	3.2–4.3	0.3	1.95	0.0511	0.34
Commute To/From School	6	2.7	2.7–3.2	2.8	2.7–3.5	0.1	3.05	0.0023	0.55
Family/Community Time	3	2.7	2–3	3	2–3.3	0.3	1.18	0.2381	0.21

* Interquartile Range 25–75%. ** Wilcoxon Signed Rank Test. *** Effect Size.

**Table 3 ijerph-19-16950-t003:** Descriptive statistics and results from the Wilcoxon Signed Rank tests for changes in stakeholder physical activity behaviours.

Domain	Items	Pre		Post		Change	Z **	*p*	r ***
		Median	IQR *	Median	IQR *				
Stakeholder Capability	16	2.9	2.6–3.3	3.4	3–3.6	0.5	3.2	0.0006	0.58
Stakeholder Opportunity	19	2.8	2.6–3.1	3.2	2.9–3.5	0.4	3	0.0018	0.57
Stakeholder Motivation	7	2.9	2.6–3.3	3.3	2.7–3.6	0.4	2.6	0.0073	0.48

* Interquartile Range 25–75%. ** Wilcoxon Signed Rank Test. *** Effect Size.

## Data Availability

The datasets used and/or analysed during this specific current study are available from the corresponding author on reasonable request.

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
