# Peer review of "Assessing the Impacts of Creating Active Schools on Organisational Culture for Physical Activity"

_ijerph, 2022, doi:10.3390/ijerph192416950_

Round 1

Reviewer 1 Report

Overall, this manuscript is very well prepared and the scientific aspects appear to be appropriate. Increased transparency about the CAS program is needed. Specifically, details can be added to demonstrate the alignment between the support provided and the variables measured for organizational and behavioral change. There are also some areas where revisions would help to improve clarity. I hope my specific comments below are helpful.

Line 47 – This statement about the ineffectiveness of attempts to increase PA during school seems to be based on research that examines naturally occurring PA opportunities/behaviors, but it should be based on intervention research.

Line 70 – Not sure it is accurate to say the CSPAP framework lacks strong theoretical underpinning from behavioral and implementation science. The framework is rooted in a social-ecological perspective and multiple theories have been used to build the evidence that informs best practices for CSPAP implementation. Further information can be found in the text by Carson and Webster (2020) – Comprehensive School Physical Activity Programs: Putting Research into Evidence-based Practice.

Line 83 – Program or framework? (If program, please explain how this is different from the framework.)

Line 88 – The purpose statement is long and a bit unclear. Please revise for clarity.

Line 89 – Program or framework?

Line 90 – “through providing” repeated – please revise.

Line 100 – Please provide more details about the implementation team – who from each organization is involved and in what ways?

Line 102 – What is the difference between designing and delivering the program?

Line 106 – The three delivery models need further description. For example, how did the differences in funding change the delivery or implementation approaches?

Line 121 – Please elaborate as to what the CAS Champions’ “external support” consisted of.

Line 130 – Change “outlines” to outlined.

Line 145 – Please provide more details about the communities of practice. For example, were these in person or online and what kind of information was shared?

Line 150 – Please provide examples of these initiatives (e.g., what kinds of trainings were provided, what did direct support for physical activity look like?).

Line 170 – Change “sheer” to sheet.

Line 181 – Change “was sent” to were sent.

Line 187 – Please give more details about the team of international researchers with expertise in school-based physical activity.

Line 189 – Please be more specific about what “reviewing and validating” involved in the first step of questionnaire development.

Line 195 – Please give examples of amendments that were made to the questionnaire, based on the feedback from the head teacher and independent academic.

Line 197 – Please give examples of amendments that were made to the questionnaire, based on the feedback from the CAS team and CAS Champions.

Lines 227-228 – There are some typos.

Line 241 – Wondering whether the Wilcoxon Signed Rank Test is appropriate, based on the data being ordinal. Isn’t the choice to use this test based instead on whether the data are normally distributed or not?

For data analysis, please add justification for not subjecting the questionnaire items to factor analysis.

Line 283 – What is meant by “at least” in this sentence?

Table 2 – Organizational change is listed twice.

Line 325 – The results showed a medium effect for the change in motivation, not a large effect.

Line 329 – Please provide an explanation as to why you think parent behavior did not change. Is there any previous literature that can help to explain this result?

Line 335 – Was any information available as to how schools used the funding to implement the CAS program?

Line 337 – Maybe change “As a result, the results…”

Line 339 – Which evidence base?

Line 340 – “and other implementation frameworks” repeated – please revise.

Line 340 – Please spell out CFIR before using the acronym.

Line 345 – Somewhat awkward sentence – please revise for clarity.

Line 349 – Please explain what is meant by “delve into the context of underlying mechanisms.”

Line 352 – Systematic or systemic?

Line 355 – Outlines or outlined?

Line 355 – Sentence is hard to follow – please revise for clarity.

Line 358 – This sentence is vague – please be more specific about what “moving beyond” previous approaches should entail.

Line 359 – Perhaps change “individualistic” to single component.

Line 363 – Before discussing why the results should be interpreted with caution, it would be useful here to explain why the observed changes in capability, opportunity and motivation are important in reference to the previous literature.

Line 370 – Not sure how the example provided here (of no changes in school leaders’ behavior) illustrates the point made in the previous sentence about the sample being potentially biased.

Line 377 – Maybe change “as this can be a barrier” to these factors can be a barrier.

Line 384 – Please elaborate on the flexible approach described here and maybe provide an example or two.

Line 388 – Change “Parents” to parents.

In the strengths and limitations section, it may be important to add that observational protocols could provide more objective data and support “groundtruthing” in relation to school practices and staff behaviors.

Line 464 – Maybe provide some examples of “broader stakeholders.”

Author Response

Thank you for taking the time to review and provide comments on this paper, we have found these useful when making revisions. The manuscript has been edited based on the feedback provided, and we feel that the paper has been strengthened through the amendments made. Please see below for a point-by-point response to each of the reviewers comments.

Overall, this manuscript is very well prepared and the scientific aspects appear to be appropriate. Increased transparency about the CAS program is needed. Specifically, details can be added to demonstrate the alignment between the support provided and the variables measured for organizational and behavioral change. There are also some areas where revisions would help to improve clarity. I hope my specific comments below are helpful.

Line 47 – This statement about the ineffectiveness of attempts to increase PA during school seems to be based on research that examines naturally occurring PA opportunities/behaviours, but it should be based on intervention research.

Thank you for the comment. We have reworded the sentence to reflect research (Ref 7) and international guidance (Ref 8 & 9) recommending whole-school approaches. (see lines 48-51) 

“It is therefore important to understand why attempts to increase children’s physical activity levels have been unsuccessful via school-based approaches when research and international guidance suggests that they are effective settings within which to intervene [7–9].

Line 70 – Not sure it is accurate to say the CSPAP framework lacks strong theoretical underpinning from behavioral and implementation science. The framework is rooted in a social-ecological perspective and multiple theories have been used to build the evidence that informs best practices for CSPAP implementation. Further information can be found in the text by Carson and Webster (2020) – Comprehensive School Physical Activity Programs: Putting Research into Evidence-based Practice.

We have altered the sentence to reflect on the original development of the models, removing theoretical perspective (which suggests no theoretical underpinning) to ensure the reader understands the point is focussed on behavioural and implementation science.  (see lines 71-73)  

“However, while behavioural theory and implementation science have been added in recent years to the aforementioned models and subsequent programmes, both were lacking within their original development [17,18].” 

Line 83 – Program or framework? (If program, please explain how this is different from the framework.)

In this instance we have used the word framework, and we have altered the following  sentence that differentiates the framework from the programme. (See Line 85-92). 

“The framework underpinned the development of the CAS programme that focuses on transforming organisational culture to promote physical activity in schools across four domains: policy, environments, stakeholders, and opportunities. The framework incorporates both the COM-B model of behaviour change that identifies three factors that must be present for behaviour change to occur [17], and the Consolidated Framework for Implementation Research to identify physical activity determinants across multiple levels of the organisation [20] (Supplementary File 1).”

Line 88 – The purpose statement is long and a bit unclear. Please revise for clarity.

We have shortened the length of the  primary purpose statement and reworded it to improve clarity (see line 93-95)  

“This study aims to evaluate the effectiveness of the CAS programme on organisational culture  for  physical activity in schools. A secondary aim is to establish the internal validity and reliability of the school-based organisational capacity questionnaire.”

Line 89 – Program or framework?

Thank you for this comment, we have kept the word programme as it is this that is being evaluated rather than the framework. However, we have included a sentence explaining the difference between the programme and the framework, as explained in the previous point (see line 85). 

Line 90 – “through providing” repeated – please revise.

This has been removed.

Line 100 – Please provide more details about the implementation team – who from each organization is involved and in what ways?

We have added the two sentences to outline the roles to support programme implementation across the different organisations. (see line 105-110) 

CAS is led by a national team (University of Bradford (UoB), Yorkshire Sport Foundation (YSF) and Bradford Institute for Health Research (BIHR)) who design and implement the programme across the UK. The programme has strategic leads for research and development (UoB) and partnership development (YSF). A national manager (YSF) leads the delivery for locality leads across a national community of practice, of which the Bradford locality is a member.”

Line 102 – What is the difference between designing and delivering the program?

We have removed the word “deliver”, please see sentence in response above. We now use design and implement. 

Line 106 – The three delivery models need further description. For example, how did the differences in funding change the delivery or implementation approaches?

We have altered the following sentence to clarify the funding does not alter the delivery model, only the in-school implementation of chosen initiatives. (see 117-120). 

“JU:MP and CP5 both supplemented the support for CAS with small grants (£4K-£10K). All schools received the same implementation support by the CAS team, with the funding providing additional resource to support the schools to implement their chosen initiatives.” 

Line 121 – Please elaborate as to what the CAS Champions’ “external support” consisted of.

We have reworded the sentence for clarity as the external support was listed after this statement but appreciate the initial wording did not make this clear for the reader. (see lines 131-134). 

“Their role was to provide external support to schools, to facilitate engagement with CAS, check and challenge during the profiling exercise, support the development and implementation of school-based initiatives and connect schools into locality-based communities of practice.”

Line 130 – Change “outlines” to outlined.

This has been amended to “outlined”. 

Line 145 – Please provide more details about the communities of practice. For example, were these in person or online and what kind of information was shared?

We have added that these were face-to-face and also provided examples at the end of the sentence to add context. (see line 155-159). 

“The CAS Champion created face-to-face communities of practice amongst their allocated schools to develop and support the implementation of shared school initiatives (e.g., integrating physical activity within teaching and learning policy, developing active travel initiatives, and outdoor physically active learning opportunities).”

Line 150 – Please provide examples of these initiatives (e.g., what kinds of trainings were provided, what did direct support for physical activity look like?).

We have expanded the sentence and provided contextualised examples after each of the key framework areas. (see 162-169) 

“The CAS Champion supported implementation of individual and collective school initiatives. The locality leadership organised termly conferences/ communities of practice (three times a year) to support the development and implementation of initiatives, these included; policy(e.g., integrating physical activity in teaching policy) or environment improvements (e.g., building an outdoor classroom), stakeholder training (e.g., physically active learning continuing professional development), and direct support for physical activity within one of the seven CAS opportunity areas for physical activity (e.g., bikeability training to support active travel).”

Line 170 – Change “sheer” to sheet.

Thank you, this has been changed to “sheet”.

Line 181 – Change “was sent” to were sent.

Thank you, we have amended this to “were sent”.

Line 187 – Please give more details about the team of international researchers with expertise in school-based physical activity.

We have removed the word “international”.  To improve clarity we have acknowledged it was the author team who were involved in the development of the questionnaire which reflects an international set of researchers. (see lines 207-211)

“First, the paper authors  who have expertise in school-based physical activity, behaviour change and implementation science reviewed the previously validated school wellness questionnaire (SWQ) [24], CAS framework [19], CAS profiling tool, COM-B model [21], and six-item COM-B questionnaire assessing effectiveness of interventions [25].”

Line 189 – Please be more specific about what “reviewing and validating” involved in the first step of questionnaire development.

We have added in the word “previously” to clarify this was a review of previously validated questionnaires.  Please see the wording above. 

Line 195 – Please give examples of amendments that were made to the questionnaire, based on the feedback from the head teacher and independent academic.

Line 197 – Please give examples of amendments that were made to the questionnaire, based on the feedback from the CAS team and CAS Champions.

We have included information on the changes which were suggested that focussed on the simplification of language within the questionnaire (see line 214-219).

“Both provided feedback on questions’ context and structure. Following amendment, the questionnaire was then trialled with the national CAS team and Bradford-based CAS Champions (n=12) before final amendments were made to the structure and question wording. Amendments across both stages of feedback specifically focused on simplifying the behavioural terminology for school stakeholders.”

Lines 227-228 – There are some typos.

Typos have been amended. (see lines 248-251)

“Individual items were included if they achieved an items-rest correlation score of α≥0.3 [27]. The questionnaire and each domain was accepted as worthy of retention if a score of α≥0.7 was achieved [26]. If individual items scored α<0.3 they were removed from analysis and the domain was  re-assessed.”

Line 241 – Wondering whether the Wilcoxon Signed Rank Test is appropriate, based on the data being ordinal. Isn’t the choice to use this test based instead on whether the data are normally distributed or not?

We have added further detail to clarify why the Wilcoxon Signed Rank Test was used. (see lines 263-266)

“As data were ordinal, non-parametric and not normally distributed, the Wilcoxon Signed Rank Test was performed to assess the difference in physical activity provision at an organisational level from when schools joined the programme to the nine-month engagement point.”

For data analysis, please add justification for not subjecting the questionnaire items to factor analysis.

Thank you for this point, we have included justification as to why factor analysis was not used (see lines 268-271).

“Factor analysis was not performed due to (a) the questionnaire within this study being developed from previously validated questionnaires, and (b) the sample size being deemed too small, and rather the reliability (Cronbach Alpha) was more suitable for this study [26,30].”

Line 283 – What is meant by “at least” in this sentence?

We have deleted the words “at least” (see lines 309-311).

“Three quarters of schools rated the provision of physical activity in PE (75%), break/ lunch (75%), events/visits: external sites (81%), commute to/from school (87%) and family/community (81%) as insufficient.”

Table 2 – Organizational change is listed twice.

We have deleted the duplicate row from the table. 

Line 325 – The results showed a medium effect for the change in motivation, not a large effect.

We have altered this sentence to reflect medium effects for Motivation (see lines 350-352). 

“Overall, there were large significant effects of CAS on Stakeholder Capability and Opportunity, and medium effects on Stakeholder Motivation to adopt and implement a whole-school approach to physical activity.”

Line 329 – Please provide an explanation as to why you think parent behavior did not change. Is there any previous literature that can help to explain this result?

We have addressed this point in lines 428-434, where the context for the parents' results are explained and linked to previous literature 

“These findings are consistent with the SWITCH programme where parents had low levels of engagement due to reduced representation within the programme [24]. Moving forward, CAS and other whole-school approaches to physical activity need to develop effective parent engagement strategies to support parents in promoting and providing physical activity opportunities throughout the school day and beyond as they are the key agents in facilitating children’s health behaviours [38]”.

Line 335 – Was any information available as to how schools used the funding to implement the CAS program?

We did not capture this detail in the current study. We have added a sentence to the limitations to reflect the need to understand the contextual factors that impact implementation, with specific reference to funding allocation between different sub-programmes. (see line 488-491) 

“In addition, future studies would benefit from a mixed-method approach to provide an understanding of how contextual factors and varied programme funding influences engagement and implementation of the CAS programme.”

Line 337 – Maybe change “As a result, the results…”

Altered the wording to consequently, (see line 365-367)  

“Consequently, the results from this study indicate that the CAS programme is broadly successful in guiding the development of opportunities for physical activity in schools.”

Line 339 – Which evidence base?

Added in the words “surrounding whole-school approaches to physical activity (see lines (367-370)

“These findings contribute to the growing evidence base surrounding whole-school approaches to physical activity, reinforcing the use of CFIR and other implementation frameworks and strategies to create organisational change in schools for physical activity provision [30].”

Line 340 – “and other implementation frameworks” repeated – please revise.

Deleted repetition 

Line 340 – Please spell out CFIR before using the acronym.

We have used the full words in the introduction and added the acronym after this (see lines 88-92) 

“The framework incorporates both the COM-B model of behaviour change that identifies three factors that must be present for behaviour change to occur [17], and the Consolidated Framework for Implementation Research (CFIR) to identify physical activity determinants across multiple levels of the organisation [20] (Supplementary File 1).”.

Line 345 – Somewhat awkward sentence – please revise for clarity.

We have reworded this sentence for clarity. (see line 373-375) 

“Results may be similar due to the SWITCH measure of organisational capacity underpinning the current questionnaire [24].”

Line 349 – Please explain what is meant by “delve into the context of underlying mechanisms.”

We have simplified this sentence to improve clarity. (see lines 376-379) 

“While initial change has been detected at nine months of CAS and implementation support, future research needs to track the longevity of change and investigate the underlying mechanisms of change and the contextual factors that likely influence these.”

Line 352 – Systematic or systemic?

We have used systemic. 

Line 355 – Outlines or outlined?

This has been amended to outlined.

Line 355 – Sentence is hard to follow – please revise for clarity.

We have simplified the sentence to (see line 386-389) 

“The mechanisms by which policy and higher system improvements lead to behaviour change at the pupil level, are through enhanced social and physical environments and improved stakeholder behaviour to adopt and implement physical activity [19,32].”

Line 358 – This sentence is vague – please be more specific about what “moving beyond” previous approaches should entail.

We have added more specific wording (see line 389-392)

“In combination with previous work [32,33], the results address the need for national guidance to recommend whole-school approaches rather than what have often been ineffective single component or multi-component interventions.”

Line 359 – Perhaps change “individualistic” to single component.

Thank you, we have made this amendment. 

Line 363 – Before discussing why the results should be interpreted with caution, it would be useful here to explain why the observed changes in capability, opportunity and motivation are important in reference to the previous literature.

We have added two sentences to outline the impact of the findings and links these to a recent meta-synthesis. (see lines 395-398)

“As seen in a recent meta-synthesis, changes in all three domains of behaviour are essential to improve physical activity delivery by teachers and wider school staff. The results of this study provide novel insights into how stakeholder behaviour can positively influence broader whole-school culture for physical activity.”

Line 370 – Not sure how the example provided here (of no changes in school leaders’ behavior) illustrates the point made in the previous sentence about the sample being potentially biased.

We have removed the sentence that references bias and have started a new paragraph  focussing on the context of the senior leaders and no change in their scores. (see lines 401-407)

“As a result, the improvements may reflect a novelty to engage in an initiative and further work is required to track changes over the longer term [35], along with actual change in pupil physical activity behaviours.

Surprisingly, changes in behaviour were not observed in school leaders.”.

Line 377 – Maybe change “as this can be a barrier” to these factors can be a barrier.

We have altered the wording as suggested (see lines 412-415) 

“This might be best achieved through qualitative work to understand the complexity and perceived compatibility and relative advantage of CAS as these factors can be a barrier to senior leaders adopting and maintaining practices at a school-wide level [4,36].”

Line 384 – Please elaborate on the flexible approach described here and maybe provide an example or two.

We have removed “flexible” and focussed on individualisation and then provided context. (see line 420-423)  

“Similar to the Finnish Schools on the Move programme, CAS empowers school staff to adopt and maintain new practices in individualised ways (e.g., selecting initiatives based on current school assets and contexts) [32,33].”

Line 388 – Change “Parents” to parents.

Thank you, this has been ammended to parents’.

In the strengths and limitations section, it may be important to add that observational protocols could provide more objective data and support “groundtruthing” in relation to school practices and staff behaviors.

Thank you for this point, we have used some of your wording to add an additional sentence to the strengths and limitations section (see lines 482-483).

“Therefore, observational studies would provide more objective data to corroborate self-report outcomes in relation to school practices and staff behaviours.”

Line 464 – Maybe provide some examples of “broader stakeholders.”

We have provided two examples (see lines 513-516) 

“To ensure a true whole-school approach, it will be essential for CAS to work with broader stakeholders within and beyond schools (e.g., parents and public health specialists) to adopt and implement physical activity.”

Reviewer 2 Report

Thank you for the opportunity to review this manuscript. I read the study with great interest. There are a few questions I have for the authors, that if addressed, may help clarify and strengthen the manuscript for readers. 

1. The study assessed organizational change using a questionnaire on the novel CAS programme. It seems like one person from the school filled out the questionnaire (i.e., the team lead). What impact does this have on assessing organizational change as a whole, without delivering the questionnaire to other stakeholders? Did the authors collect expectancy beliefs of these team leads who may be susceptible to priming effects (anticipating a change in outcomes assessed by the survey given their participation in the intervention)? No research study is perfect so if the authors are unable to address these limitations, they should at the very least be acknowledged in the discussion. In the discussion, it seems like multiple stakeholders were assessed, but looking at the tables and reading the methods, this did not come through strongly. It might be worthwhile revisiting these sections to make sure readers understand who responded to the surveys that are presented in Table 2 and 3.

2. Another limitation and future direction is assessing organization change based on sparse sampling points (pre/post). What limitations does this play on our understanding when there is lots that organizationally may have changed between 9-months of implementation (i.e., staff turnover, new leadership, weather patterns influencing activity levels, being a part of an intervention as a teacher and expecting we should "show" something changes)? There is lots that can happen to influence the observed changes and basing our conclusion that the program is the sole cause of these changes has limits. Future work can address this by examining within-day and daily level changes/fluctuations in organizational change from the program (i.e., ecological momentary assessment, passive sensor studies). This would help get at causal mechanisms as well, which cannot be taken from the current data (i.e., that the program caused changes between 9 months cannot be concluded; we can simply say things are different from 9 months ago but we can't conclude it is because of the program).

3. The authors acknowledge some of these limitations in towards the end of their discussion, but it is much more than longitudinal change and persistence of these effects that needs to be examined; it is also denser sampling methods on finer timescales that are needed in addition to complex systems and network science tools that can examine the interactions across multiple levels and systems. I implore the authors to consider these future directions. But to use these statistical techniques, there needs to be intensive longitudinal measurement time series. 

Author Response

Thank you for taking the time to review and provide comments on this paper, we have found these useful when making revisions. The manuscript has been edited based on the feedback provided, and we feel that the paper has been strengthened through the amendments made. Please see below for a point-by-point response to each of the reviewers comments.

Thank you for the opportunity to review this manuscript. I read the study with great interest. There are a few questions I have for the authors, that if addressed, may help clarify and strengthen the manuscript for readers. 

  1. The study assessed organizational change using a questionnaire on the novel CAS programme. It seems like one person from the school filled out the questionnaire (i.e., the team lead). What impact does this have on assessing organizational change as a whole, without delivering the questionnaire to other stakeholders? Did the authors collect expectancy beliefs of these team leads who may be susceptible to priming effects (anticipating a change in outcomes assessed by the survey given their participation in the intervention)? No research study is perfect so if the authors are unable to address these limitations, they should at the very least be acknowledged in the discussion. A limitation of the approach is that the questionnaires were completed by one stakeholder to reflect the whole organisation. In addition, the CAS lead, a stakeholder with knowledge of the programme, completed this questionnaire leaving responses open to bias. This is commonly a limiting factor within self-report research.

Thank you for this comment, we have included a sentence within the strengths and limitations section addressing this as a limitation within the study and how it can be developed moving forwards (see lines 480-483).

“Reliance on one individual, who may be susceptible to priming beliefs, to assess overall organisational changes reduces the validity of findings. Therefore, observational studies would provide more objective data to corroborate self-report outcomes in relation to school practices and staff behaviours”.

In the discussion, it seems like multiple stakeholders were assessed, but looking at the tables and reading the methods, this did not come through strongly. It might be worthwhile revisiting these sections to make sure readers understand who responded to the surveys that are presented in Table 2 and 3.

We have clarified in the methodology that the in-school CAS lead completed the questionnaire on behalf of the school and stakeholders within (see lines 193-195).

“School readiness and organisational capacity for physical activity were assessed over a nine-month period via a questionnaire which the in-school CAS leads completed on behalf of their school and stakeholders.”

  1. Another limitation and future direction is assessing organization change based on sparse sampling points (pre/post). What limitations does this play on our understanding when there is lots that organizationally may have changed between 9-months of implementation (i.e., staff turnover, new leadership, weather patterns influencing activity levels, being a part of an intervention as a teacher and expecting we should "show" something changes)? There is lots that can happen to influence the observed changes and basing our conclusion that the program is the sole cause of these changes has limits. Future work can address this by examining within-day and daily level changes/fluctuations in organizational change from the program (i.e., ecological momentary assessment, passive sensor studies). This would help get at causal mechanisms as well, which cannot be taken from the current data (i.e., that the program caused changes between 9 months cannot be concluded; we can simply say things are different from 9 months ago but we can't conclude it is because of the program).

We have included this within the strengths and limitations section, recognising the need for observational and mixed methods approaches to corroborate findings and increase understanding (see lines 482-483 and 488-491)

“Therefore, observational studies would provide more objective data to corroborate self-report outcomes in relation to school practices and staff behaviours.”

“In addition, future studies using a mixed-method design and a broader range of stakeholders would improve understanding of how contextual factors and varied programme funding influence engagement and implementation of the CAS programme.”

  1. The authors acknowledge some of these limitations in towards the end of their discussion, but it is much more than longitudinal change and persistence of these effects that needs to be examined; it is also denser sampling methods on finer timescales that are needed in addition to complex systems and network science tools that can examine the interactions across multiple levels and systems. I implore the authors to consider these future directions. But to use these statistical techniques, there needs to be intensive longitudinal measurement time series. 

Thank you, this is a very interesting point. Using some of the advice above we have included this within our future directions (see lines 462-466)

“This requires a longitudinal follow up across the length of time predicted for the transformation of school culture (years vs months). However, denser sampling methods on finer timescales are also warranted to examine the causal mechanisms behind change and the interactions across the multiple levels of the school system.”